# Anisotropic Nucleation, Growth and Ripening under Stirring—A Phenomenological Model

**DOI:** 10.3390/e22111254

**Published:** 2020-11-04

**Authors:** Andriy Gusak, Yaroslav Huriev, Jürn W. P. Schmelzer

**Affiliations:** 1Department of Physics, Cherkasy National University, 18000 Cherkasy, Ukraine; yaroslavhuriev@gmail.com; 2Institute of Physics, University of Rostock, Albert-Einstein-Strasse 23-25, 18059 Rostock, Germany; juern-w.schmelzer@uni-rostock.de

**Keywords:** kinetics, nucleation, growth, ripening, anisotropy, driven system, stirring, ballistic events, fibers, crystallization

## Abstract

The anisotropic formation of elongated metal-oxide aggregates in water under intensive stirring is analyzed. It is treated in terms of anisotropic ballistically mediated aggregation kinetics in open systems. The basic kinetic equations describing the stages of homogeneous nucleation, independent growth, and ripening of the aggregates are formulated for the open system under the external influence with the stirring intensity as the main parameter governing the process. The most significant elongation of the aggregates is shown to evolve at the ripening stage.

## 1. Introduction

Some years ago, a new method of production of rod-like and belt-like metal oxide structures (TiO_2_, V_2_O_5_) was suggested involving intensive stirring of salted water with initially more or less equiaxially shaped oxide powder particles at elevated or even at room temperatures [1,2]. This new method connected the problem of anisotropic nucleation, growth, and ripening [3,4,5,6] with the problem of phase and structural transformations in open driven systems [7,8,9,10,11,12,13,14,15,16,17]. In order to develop a theoretical interpretation of these interconnected phenomena, the first very simplified models were advanced recently in [18]. Among other steps, ballistic terms (proportional to stirring intensity) were added to the kinetic equations of anisotropic precipitate growth and shrinkage following general ideas formulated long ago by G. Martin et al. [7,8,9,10].

In the present paper, we develop a more systematic approach to the general problem of description of anisotropic nucleation/growth/coarsening under ballistic influence. Taking into account such process conditions, new phenomena may occur not found at standard process conditions. In particular, the coarsening (Ostwald ripening) stage of phase/structural transformation in closed systems at constant pressure and temperature is driven by the tendency to decrease the total surface energy of the aggregates keeping the total volume of the newly evolving phase constant. This is a consequence of the thermodynamic evolution criterion implying for such conditions a decrease in the Gibbs free energy of the system. For these boundary conditions specified, any evolution process has to obey the mentioned thermodynamic requirements. 

In contrast, in open systems, the situation is not that straightforward, because external factors acting on the system under consideration may compete with this tendency. For example, in the so-called Flux Driven Ripening (FDR)—the ripening of Cu_6_Sn_5_ scallops in the reaction zone between solid copper and liquid tin-based solder—the total surface remains almost constant instead of decreasing. The decrease in the Gibbs free energy is not anymore the basic criterion of evolution and has to be replaced by other evolutions criteria accounting for externally induced fluxes and chemical reactions [15]. In our case of nucleation and growth under stirring, the latter mentioned factors may retain the supersaturation at finite values, so it will not tend to zero as is the case in segregation processes in solutions in closed systems [12]. As a consequence, common regularities of ripening in closed systems may be violated. Such new unexpected in advance effects may occur, in particular, in cases if the kinetics of attachment and detachment differs at different facets of anisotropic particles. The analysis of this problem is the aim of the present paper.

The paper is structured as follows. In Section 2, we formulate the basic kinetic equations employed for the description of growth and shrinkage under stirring. These equations are written within the framework of the linear approximation of non-equilibrium thermodynamics (Onsager approach) but with the addition of ballistic terms (going beyond standard non-equilibrium thermodynamics). In Section 3.1, we discuss anisotropic nucleation in open systems under stirring. According to classical nucleation theory (CNT), nucleation implies overcoming of clusters of the newly evolving phase of the nucleation barrier following a path of evolution passing the barrier somewhere in the vicinity of the saddle-point of the Gibbs free energy. This process is generated by random attachments and detachments of monomers (diffusion in cluster size space) additionally to the drift factors (shrinkage at subcritical sizes and growth at supercritical sizes). In this paper, we mainly analyze the influence of stirring on the drift terms in the equations governing nucleation and leave for future the possible modifications in the noise terms. In Section 3.2, we analyze briefly the evolution of sizes and of aspect ratios for supercritical particles under stirring at almost constant supersaturation (the growth stage). In Section 3.3 we will develop a theoretical approach to the description of the ripening stage in open systems advancing the well-known Lifshitz–Slezov–Wagner (LSW) theory valid for coarsening in closed systems. To obtain an impression on typical time scales for the processes under consideration, one should keep in mind that the typical real time to obtain fibers or nanobelts with mean length up to several tens of microns and lateral size up to one micron, under stirring with rotation frequency up to 1000 rotations per minute, is up to 300,000 s. A discussion of the results and possible future developments completes the paper.

## 2. Methods (Basic Equations)

In [18], we derived simple kinetic equations allowing one to describe, in the first approximation, the anisotropic growth or shrinkage of nanobelts. These kinetic equations consist of a set of three equations containing parameters describing the surface tension, kinetic coefficients for attachment/detachment at quasi-equilibrium conditions, and ballistic detachment frequencies (per unit area) for the different directions (three facets). Here, we present a more advanced method of analysis allowing one to treat the phenomena in more detail. In order to proceed, we assume a definite shape of the evolving aggregates in line with the experimental evidence discussed in the introduction. We take as a model system simple fiber (cylindrical) structures which may be described by two different size parameters, length *l* and diameter *D*, and two different surface tensions $\gamma_{R}$ (side tension) and $\gamma_{l}$ (top tension). Alternatively, one could consider hexagonal prismatic structures [6], the methods developed here are applicable directly also to such and other similar cases. 

In the present analysis, we will consider only “free-standing” rods (with diameter *D* and length *l*), though, according to [2], they may nucleate and then grow in one direction at the base of some facet of an initially more or less equiaxial precipitate. Simple considerations, treating the chemical potential μ of a given facet via an additional contribution to the Gibbs free energy per atom due to the adding of a thin layer to this facet, result in the following form of the Gibbs–Thomson relations for aggregates of cylindrical shape:(1)$$\left\{ \begin{array}{l} \mu_{D}^{cryst}=\mu^{side}={\frac{\partial G}{\partial N}|}_{dD}=\frac{\mu_{}^{cryst}\cdot \frac{\pi DldD/2}{\Omega }+\gamma_{D}l\cdot \pi dD+2\gamma_{l}d\left(\pi D^{2}/4\right)}{\frac{\pi DldD/2}{\Omega }}=\\ =\mu^{bulk}+\frac{2\gamma_{D}\Omega }{D}+\frac{2\gamma_{l}\Omega }{l}\\ \mu_{l}^{cryst}=\mu^{top}={\frac{\partial G}{\partial N}|}_{dl}=\frac{\mu_{}^{cryst}\cdot \frac{\pi D^{2}/4\cdot dl}{\Omega }+\gamma_{D}dl\cdot \pi D}{\frac{\pi D^{2}/4\cdot dl}{\Omega }}=\mu^{bulk}+\frac{4\gamma_{D}\Omega }{D} \end{array}\right.$$

Here *G* is the Gibbs free energy, *N* is the number of monomers in the rod of diameter *D* and length *l*, Ω is the volume per monomer (atom or molecule), In the case of thermodynamic equilibrium (equal chemical potentials at all facets), one obtains Wulff’s rule $\frac{\gamma_{D}}{D}=\frac{\gamma_{l}}{l}$ as a special case from these relations. Evidently, surface tensions at different facets cannot differ by several orders of magnitude. Thus, the formation of fibers and belts is a process possible only at substantially non-equilibrium conditions. 

Following the ideas of ballistic growth in driven systems [7,8,9,10], one may write down the main phenomenological kinetic equations for a “free-standing” cylinder in solution under stirring [18]: (2)$$\left\{ \begin{array}{l} \frac{dD}{dt}=-2L_{D}{\frac{\partial G}{\partial N^{cryst}}|}_{l}-2U_{D}=-2L_{D}{\frac{\partial \left(G^{liq}+G^{cryst}+G^{surface}\right)}{\partial N^{cryst}}|}_{l}-2U_{D}=\\ =2L_{D}\left(\mu^{liq}-\mu_{D}^{cryst}\right)-2U_{D}\\ \frac{dl}{dt}=-2L_{l}{\frac{\partial G}{\partial N^{cryst}}|}_{D}-2U_{l}=-2L_{l}{\frac{\partial \left(G^{liq}+G^{cryst}+G^{surface}\right)}{\partial N^{cryst}}|}_{D}-2U_{l}=\\ =2L_{l}\left(\mu^{liq}-\mu_{l}^{cryst}\right)-2U_{l} \end{array}\right.$$

In such an approach, growth and shrinkage are interface-controlled processes since stirring makes effective diffusion practically instantaneous. Here, $L_{D},L_{l}$ are the kinetic (actually, Onsager) coefficients for thermally activated atomic detachments (in case of undersaturation) and attachments (in case of supersaturation) from the side surface and top surface, respectively. A simplified atomistic derivation of these coefficients and of Equation (2) can be found in [18]. $U_{D},U_{l}$ are the velocities of ballistic (athermal) erosion (due to intensive stirring) of the side and top surfaces, respectively. Factors 2 in Equation (2) account for growth/shrinkage of the free-standing wire from both sides of rod length and at opposite points of diameter. 

In principle, the kinetic coefficients $L_{D},L_{l}$ may depend also on the stirring intensity. Such an effect can be described, for example, via the effective temperature, *T^ef^*, introduced by Martin [7,8]
$$T^{ef}=T\cdot \left(1+\Delta \right)=T\cdot \left(1+\alpha J^{stirring}\right),\;L∼\exp\left(-\;\frac{Q}{kT\left(1+\alpha J^{stirring}\right)}\right)$$

Here Q is the activation energy of detachment and k is the Boltzmann constant. The parameter Δ as introduced in [7] (where the ballistic effect was caused by irradiation) was just the ratio of irradiation-induced diffusivity and temperature-induced diffusivity. In our case. the value $\Delta =\alpha J^{stirring}$ should depend on the ratio of stirring-induced detachments and temperature-induced detachments. The resulting nonlinearities (typical in such an approach) lead to a more complex treatment, which will be analyzed elsewhere. In the present paper, we treat $L_{D},L_{l}$ as constants.

The chemical potential of the solute in the liquid is determined by the concentration of its atoms per unit volume, *C*. We will use for the description the dimensionless ratio, *x*, of the concentration, *C*, of the solute in the liquid and of its concentration, *C*^cryst^, in the crystalline state, i.e., $x\equiv \frac{C}{C^{cryst}}$. The solubility of oxides in water is commonly not high, so that we may use the approximation of the ideal solution. Moreover, we will consider (so far) the supersaturation $\Delta \equiv x-x_{eq}$ as small ($\Delta /x_{eq}<<1$). Then, we may write
(3)$$\mu^{liq}-\mu^{bulk}=kT\ln\frac{x}{x_{eq}}=kT\ln\frac{x_{eq}+\Delta }{x_{eq}}=kT\ln\left(1+\frac{\Delta }{x_{eq}}\right)\approx \frac{kT}{x_{eq}}\Delta$$

So,
(4)$$\left\{ \begin{array}{l} \frac{dD}{dt}=2L_{D}\left(\frac{kT}{x_{eq}}\Delta -\frac{2\gamma_{D}\Omega }{D}-\frac{2\gamma_{l}\Omega }{l}\right)-2U_{D}=\\ =\frac{2L_{D}kT}{x_{eq}}\left(\Delta -\frac{x_{eq}\cdot 2\gamma_{D}\Omega /kT}{D}-\frac{x_{eq}\cdot 2\gamma_{l}\Omega /kT}{l}-\frac{x_{eq}U_{D}}{kTL_{D}}\right)\\ \frac{dl}{dt}=\frac{2L_{l}kT}{x_{eq}}\left(\Delta -2\frac{x_{eq}\cdot 2\gamma_{D}\Omega /kT}{D}-\frac{x_{eq}U_{l}}{kTL_{l}}\right) \end{array}\right.$$

In general, supersaturation is correlated with the size distribution of the evolving aggregates by the mass conservation law [12]:(5)$$\begin{array}{l} \left(x_{eq}+\Delta \left(t\right)\right)\left(1-\frac{\sum_{i}^{N\left(t\right)}\frac{\pi }{4}D_{i}^{2}l_{i}}{V^{tot}}\right)+\frac{\sum_{i}^{N\left(t\right)}\frac{\pi }{4}D_{i}^{2}l_{i}}{V^{tot}}=\\ =\left(x_{eq}+\Delta \left(t=0\right)\right)\left(1-\frac{\sum_{i}^{N0}\frac{\pi }{4}D_{0i}^{2}l_{0i}}{V^{tot}}\right)+\frac{\sum_{i}^{N0}\frac{\pi }{4}D_{0i}^{2}l_{0i}}{V^{tot}} \end{array}$$

These equations contain many parameters with dimensions. In order to simplify the theoretical analysis, we now introduce a characteristic length, and then, using it, dimensionless sizes, time, and dimensionless ballistic terms via:(6)$$\begin{array}{l} \lambda_{0}=x_{eq}\frac{2\sqrt{\gamma_{D}\gamma_{L}}\Omega }{kT},\;\lambda_{D}=x_{eq}\frac{2\gamma_{D}\Omega }{kT}=\sqrt{\frac{\gamma_{D}}{\gamma_{L}}}\lambda_{0},\;\lambda_{l}=x_{eq}\frac{2\gamma_{l}\Omega }{kT}=\sqrt{\frac{\gamma_{L}}{\gamma_{D}}}\lambda_{0},\;\\ r^{therm}\equiv \sqrt{\frac{\gamma_{D}}{\gamma_{L}}},\;r^{kinet}\equiv \sqrt{\frac{L_{D}}{L_{l}}},\;r^{bal}\equiv \sqrt{\frac{U_{D}}{U_{l}}},\;J\equiv \frac{x_{eq}\sqrt{U_{D}U_{l}}}{kT\sqrt{L_{D}L_{l}}}\\ \tau \equiv \sqrt{L_{D}L_{l}}\frac{2kT}{\lambda_{0}x_{eq}}t,\;\rho \equiv \frac{D}{\lambda_{0}},\;z\equiv \frac{l}{\lambda_{0}} \end{array}$$

As the result, we obtain the dimensionless version of the kinetic equations in the form:(7)$$\begin{array}{l} \frac{d\rho }{d\tau }=r^{kinet}\left(\Delta -\frac{r^{therm}}{\rho }-\frac{1/r^{therm}}{z}-\frac{r^{bal}}{r^{kinet}}J\right)\\ \frac{dz}{d\tau }=\frac{1}{r^{kinet}}\left(\Delta -\frac{2r^{therm}}{\rho }-\frac{r^{kinet}}{r^{bal}}J\right) \end{array}$$

Let us roughly estimate the main characteristic parameters: The Onsager coefficients *L* for the proportionality factors between velocity and excess chemical potential, may be evaluated (very roughly) from the Nernst–Einstein relation, the width “delta” of crystal/liquid interface (a few Angstroems), and from the analogue of diffusion coefficient for crossing the interface:$$\begin{array}{l} V=\frac{Diffusivity}{kT}F=\frac{Diffusivity}{kT}\frac{\mu^{liq}-\mu^{bulk}}{\delta }=L\Delta \mu \Rightarrow \\ L∼\frac{Diffusivity}{kT\delta }∼\frac{\nu_{D}\delta }{kT}\exp\left(-\frac{Q}{kT}\right)∼\\ ∼\frac{10^{12}10^{-10}}{0.4\cdot 10^{-20}}\exp\left(-\frac{10^{-19}}{0.4\cdot 10^{-20}}\right)∼3.5\cdot 10^{11}\frac{m}{s\cdot Joule}\\ \lambda_{0}=x_{eq}\frac{2\sqrt{\gamma_{D}\gamma_{L}}\Omega }{kT}\;∼\;2\cdot 10^{-2}\frac{2\cdot 1\frac{J}{m^{2}}10^{-29}m^{3}}{4\cdot 10^{-21}J}∼10^{-10}m,\;\\ \;J\equiv \frac{x_{eq}\sqrt{U_{D}U_{l}}}{kT\sqrt{L_{D}L_{l}}}∼\frac{10^{-2}\cdot 3\cdot 10^{-10}\frac{m}{s}}{4\cdot 10^{-21}Joule\cdot 3\cdot 10^{11}\frac{m}{Joule\cdot s}}∼2\cdot 10^{-3} \end{array}$$

We will see below that in the case of stirring the inequality $\frac{r^{bal}}{r^{kinet}}>1$ implies the formation of elongated particles, and $\frac{r^{bal}}{r^{kinet}}<1$ corresponds to the formation of disc-like particles.

The conservation law is given in such notations via:(8)$$\begin{array}{l} \left(x_{eq}+\Delta \left(t\right)\right)\left(1-\frac{\sum_{i}^{N\left(t\right)}\frac{\pi }{4}\rho_{i}^{2}z_{i}}{V^{tot}/\lambda_{0}^{3}}\right)+\frac{\sum_{i}^{N\left(t\right)}\frac{\pi }{4}\rho_{i}^{2}z_{i}}{V^{tot}/\lambda_{0}^{3}}=\\ =\left(x_{eq}+\Delta \left(t=0\right)\right)\left(1-\frac{\sum_{i}^{N0}\frac{\pi }{4}\rho_{0i}^{2}z_{0i}}{V^{tot}/\lambda_{0}^{3}}\right)+\frac{\sum_{i}^{N0}\frac{\pi }{4}\rho_{0i}^{2}z_{0i}}{V^{tot}/\lambda_{0}^{3}} \end{array}$$

Equation (7) will be used for the description of the drift terms in the kinetic equations for the description of nucleation and of the size and shape evolution under stirring at the growth stage. The set of Equations (7) and (8) will be employed in the treatment of ripening under stirring. 

## 3. Results

### 3.1. Peculiarities of Anisotropic Nucleation

#### 3.1.1. Some Introductory Comments

Nucleation is caused by the random walk of the new phase clusters in size space (stochastic term), but the critical size of the aggregates is determined by the drift term in the relations describing nucleation. Here we concentrate the attention on the drift term in order to describe some of the essential characteristics of anisotropic nucleation. Here we concentrate the attention on the drift term in order to describe some of the essential characteristics of anisotropic nucleation. A detailed Fokker–Planck approach will be advanced after acquiring and analysis of new experimental information about heterogeneous nucleation at the facets of the powder particles. 

In CNT, the drift in size space (the difference between the attachment and detachment frequencies) is proportional to the derivative of the Gibbs free energy over cluster size (in analogy to the Nernst–Einstein relation in the size space): $$\Delta \nu \equiv \nu_{+}-\nu_{-}=-\frac{\left(\nu_{+}+\nu_{-}\right)/2}{kT}\frac{\partial \Delta G}{\partial N_{cluster}}=-\frac{\bar{\nu }}{kT}\frac{\partial \Delta G}{\partial N_{cluster}}$$

Therefore, in CNT, the thermodynamic criterion for the determination of the critical size (zero derivatives over size which corresponds to the maximum (or, in general, to a saddle point) of the nucleation barrier) coincides with the kinetic criterion (zero drift). In open systems, this coincidence may not be retained [19]. Of course, in cases when the evolution path is determined by more than one parameter (say, two sizes, or size and composition within the cluster) the situation is not that straightforward.

In the present section, we treat supersaturation as fixed and not perturbed by the nucleation of a single nucleus. In the anisotropic case, Nernst–Einstein-type relations should be written separately for each facet. In our case, we arrive at
$$\begin{array}{l} \Delta \nu_{D}=-\frac{{\bar{\nu }}_{D}}{kT}{\frac{\partial \Delta G}{\partial N_{cluster}}|}_{l}+\Delta \nu_{D}^{bal}=-\frac{{\bar{\nu }}_{D}}{kT}\left(\mu^{liq}-\mu_{D}^{cryst}\right)+\Delta \nu_{D}^{bal}\\ \Delta \nu_{l}=-\frac{{\bar{\nu }}_{l}}{kT}{\frac{\partial \Delta G}{\partial N_{cluster}}|}_{D}+\Delta \nu_{l}^{bal}=-\frac{{\bar{\nu }}_{l}}{kT}\left(\mu^{liq}-\mu_{l}^{cryst}\right)+\Delta \nu_{l}^{bal} \end{array}$$
Obviously, in terms of mean growth/shrinkage velocities, these equations for mean values coincide with Equations (2)–(7). 

#### 3.1.2. Anisotropic Nucleation without Stirring

For further analysis, we introduce the new variables
$$\xi \equiv \frac{1}{\Delta }\frac{r^{therm}}{\rho },\;\eta \equiv \frac{1}{\Delta }\frac{1/r^{therm}}{z},$$
so those kinetic equations without stirring are now obtained by
(9)$$\frac{d\rho }{d\tau }=r^{kinet}\Delta \left(1-\xi -\eta \right),\;\frac{dz}{d\tau }=\frac{1}{r^{kinet}}\Delta \left(1-2\xi \right)$$
As mentioned above, these kinetic equations represent drift terms in nucleation.

The critical state (both time derivatives are equal to zero) is determined via the relations:(10)$$\begin{array}{l} \frac{dD}{dt}=0\;\Leftrightarrow \xi +\eta =1\\ \frac{dl}{dt}=0\;\Leftrightarrow \;\xi =\frac{1}{2} \end{array}\}\Rightarrow \xi^{crit}=\frac{1}{2},\;\eta^{crit}=\frac{1}{2}\Rightarrow \frac{r^{therm}}{\rho^{crit}}=\frac{1/r^{therm}}{z^{crit}}\Rightarrow \frac{\rho^{crit}}{z^{crit}}={\left(r^{therm}\right)}^{2}=\frac{\gamma_{D}}{\gamma_{l}}$$

Equality $\frac{D^{crit}}{l^{crit}}=\frac{\gamma_{D}}{\gamma_{l}}$ is the well-known Wulff’s rule determining the equilibrium shape of an aggregate in a closed system. Let us now analyze at which conditions aggregates of the newly evolving phase will either grow or disappear, again.

The condition of guaranteed further shrinking, regardless of the values of the kinetic coefficients (we will call the respective states absolutely unstable or absolutely subcritical), is given by:(11)$$\begin{array}{l} \frac{dD}{dt}<0\Leftrightarrow \xi +\eta >1\\ \frac{dl}{dt}<0\;\Leftrightarrow \;\xi >\frac{1}{2} \end{array}$$

These conditions hold in region I in Figure 1. In this case, both size parameters, D and l, decrease, so that both parameters ξ+η and ξ grow. Consequently, both inequalities hold until the complete disappearance of such subcritical particles.

The condition of guaranteed further growth, regardless of the values of the kinetic coefficients (we will call the respective states absolutely stable or absolutely overcritical), is given by:(12)$$\begin{array}{l} \frac{dD}{dt}>0\Leftrightarrow \xi +\eta <1\\ \frac{dl}{dt}>0\Leftrightarrow \;\xi <\frac{1}{2} \end{array}$$

These conditions hold in region II in Figure 1. In this case, both size parameters, D and l, grow, so that both ξ+η and ξ decrease. Consequently, both inequalities will hold as long as supersaturation is kept constant (which means that we consider growth processes sufficiently far away from the ripening stage).

Regions III and IV shown in Figure 1 contain states which may lead either to growth or to shrinkage in dependence on the ratio of kinetic coefficients:Region III:
(13)$$\begin{array}{l} \frac{dD}{dt}>0\Leftrightarrow \xi +\eta <1\\ \frac{dl}{dt}<0\Leftrightarrow \;\xi >\frac{1}{2} \end{array}$$

Region IV:

(14)$$\begin{array}{l} \frac{dD}{dt}<0\Leftrightarrow \xi +\eta >1\\ \frac{dl}{dt}>0\;\Leftrightarrow \;\xi <\frac{1}{2} \end{array}$$

In such cases, both thermodynamic and kinetic factors determine the direction of the further evolution of the aggregates.

To describe in more detail the origin of such type of behavior, we have now a close look at the shape of the surface, ΔG(ρ,z), of Gibbs free energy change in nucleation. For that purpose, we express it as a function of two size parameters and determine the first- and second-order derivatives. We obtain:(15)$$\begin{array}{l} \;\frac{\partial G}{\partial \rho }={\frac{\partial G}{\partial N_{s}}|}_{\rho }\frac{\frac{1}{\Omega }\pi Dl\cdot \frac{dD}{2}}{d\rho }=\frac{\lambda_{0}^{3}\pi }{2\Omega }\rho z{\frac{\partial G}{\partial N_{s}}|}_{\rho }=-\frac{\lambda_{0}^{3}\pi }{2\Omega }\rho z\left(\frac{kT}{x_{eq}}\Delta -\frac{2\gamma_{D}\Omega }{D}-\frac{2\gamma_{l}\Omega }{l}\right)=\\ =-\frac{\lambda_{0}^{3}\pi kT}{2\Omega x_{eq}}\rho z\left(\Delta -\frac{r^{therm}}{\rho }-\frac{1/r^{therm}}{z}\right)=-\varepsilon_{0}\rho z\left(\Delta -\frac{r^{therm}}{\rho }-\frac{1/r^{therm}}{z}\right) \end{array}$$
where
(16)$$\begin{array}{l} \varepsilon_{0}\equiv \frac{\lambda_{0}^{3}\pi kT}{2\Omega x_{eq}}\\ \;\frac{\partial G}{\partial z}={\frac{\partial G}{\partial N_{s}}|}_{z}\frac{\frac{1}{\Omega }\pi D^{2}/4\cdot dl}{dz}=\frac{\lambda_{0}^{3}\pi }{4\Omega }\rho^{2}{\frac{\partial G}{\partial N_{s}}|}_{z}=-\frac{\lambda_{0}^{3}\pi }{4\Omega }\rho^{2}\left(\frac{kT}{x_{eq}}\Delta -\frac{4\gamma_{D}\Omega }{D}\right)=\\ =-\frac{\lambda_{0}^{3}\pi kT}{4\Omega x_{eq}}\rho^{2}\left(\Delta -\frac{2r^{therm}}{\rho }\right)=-\frac{\varepsilon_{0}}{2}\rho^{2}\left(\Delta -\frac{2r^{therm}}{\rho }\right) \end{array}$$

The second-order derivatives read:(17)$$\begin{array}{l} {\frac{\partial^{2}G}{\partial \rho^{2}}|}_{st}=\varepsilon_{0}\left(\Delta +\frac{r^{therm}}{\rho }-\frac{1/r^{therm}}{z}\right)\\ \frac{\partial^{2}G}{\partial z^{2}}=0\\ \frac{\partial^{2}G}{\partial \rho \partial z}=\frac{\partial^{2}G}{\partial z\partial \rho }=\varepsilon_{0}\left(\rho \cdot \Delta -r^{therm}\right) \end{array}$$

Simple algebraic transformations (employing the results for first and second-order derivatives and the condition of zero value of the Gibbs free energy change at zero sizes) leads to the following expression for the Gibbs free energy surface:(18)$$\Delta G=\varepsilon_{0}\cdot \left\{\frac{\rho^{2}\left(1-r^{therm}z\cdot \Delta \right)}{2r^{therm}}+2r^{therm}\rho z\right\}$$

Nucleation barrier (value of Gibbs free energy at the stationary (critical) point coinciding with the saddle-point) is given by
(19)$$\Delta G^{*}=G\left(\rho =\rho^{st}=\frac{2r^{therm}}{\Delta },z=z^{st}=\frac{2/r^{therm}}{\Delta }\;\right)=\varepsilon_{0}\cdot \frac{2r^{therm}}{\Delta^{2}}$$

At that, $\frac{\rho }{\rho^{st}}=\frac{\rho \Delta }{2r^{therm}}=\frac{1}{2\xi },\;\frac{z}{z^{st}}=\frac{zr^{therm}\Delta }{2}=\frac{1}{2\eta }$.

Dimensionless Gibbs free energy as a function of reduced sizes now is:(20)$$\frac{\Delta G}{\Delta G^{*}}={\left(\frac{\rho }{\rho^{st}}\right)}^{2}\left(1-2\frac{z}{z^{st}}\right)+2\frac{\rho }{\rho^{st}}\frac{z}{z^{st}}$$

This function is illustrated at Figure 2.

The stationary point is given by
$$\frac{d\rho }{dt}=0,\frac{dz}{dt}=0\Leftrightarrow \rho^{st}=\frac{2r^{therm}}{\Delta },z^{st}=\frac{2/r^{therm}}{\Delta }\;(\xi^{st}=1/2,\;\eta^{st}=1/2).$$

It corresponds to a saddle-point of the Gibbs free energy surface. Indeed, a saddle point has the feature that both first-order derivatives are equal to zero, one of the second-order derivatives (along the optimal path) is negative and the other one is positive (along the line perpendicular to it). 

The determinant of the matrix of second-order derivatives has a negative value
(21)$$\det\left(\begin{array}{cc} {\frac{\partial^{2}G}{\partial \rho^{2}}|}_{st} & {\frac{\partial^{2}G}{\partial \rho \partial z}|}_{st}\\ {\frac{\partial^{2}G}{\partial z\partial \rho }|}_{st} & {\frac{\partial^{2}G}{\partial z^{2}}|}_{st} \end{array}\right)=\det\left(\begin{array}{cc} 2\varepsilon_{0}/r^{therm} & \varepsilon_{0}r^{therm}\\ \varepsilon_{0}r^{therm} & 0 \end{array}\right)=-{\left(\varepsilon_{0}r^{therm}\right)}^{2}<0$$

It means that the matrix of second-order derivatives has one positive and one negative eigenvalue
(22)$$G_{1,2}^{\prime \prime }=\frac{1}{r^{therm}}\pm \sqrt{{\left(\frac{1}{r^{therm}}\right)}^{2}+{\left(r^{therm}\right)}^{2}},\;G_{1}^{\prime \prime }>0,\;G_{2}^{\prime \prime }<0.$$

Regions III and IV exhibit a very important peculiarity of anisotropic nucleation, which may occur, as shown, even in closed systems. In some cases, the chemical potential in the liquid can be higher than that at one facet and simultaneously lower than that at another facet. In this case, the nucleus/embryo should have the tendency to grow in one direction and dissolve in another direction (if one considers only drift terms in the kinetic equations and neglects random walk in the size space). Of course, as shown explicitly above, in all cases the drift terms may lead only to a decrease in Gibbs free energy. Indeed, utilizing the above relations, we obtain, for the change of the Gibbs free energy dependant on time, the following result:$$\frac{dG}{dt}=\frac{\partial G}{\partial D}\frac{dD}{dt}+\frac{\partial G}{\partial l}\frac{dl}{d\tau }=-2L_{D}{\left(\frac{\partial G}{\partial D}\right)}^{2}-2L_{l}{\left(\frac{\partial G}{\partial l}\right)}^{2}<0$$

Derivative over dimensionless time is similarly derived by:(23)$$\begin{array}{l} \;\frac{dG}{d\tau }=\frac{\partial G}{\partial \rho }\frac{d\rho }{d\tau }+\frac{\partial G}{\partial z}\frac{dz}{d\tau }=-\varepsilon_{0}\rho z\left(\Delta -\frac{r^{therm}}{\rho }-\frac{1/r^{therm}}{z}\right)\cdot r^{kinet}\left(\Delta -\frac{r^{therm}}{\rho }-\frac{1/r^{therm}}{z}\right)+\\ -\frac{\varepsilon_{0}}{2}\rho^{2}\left(\Delta -\frac{2r^{therm}}{\rho }\right)\frac{1}{r^{kinet}}\left(\Delta -\frac{2r^{therm}}{\rho }\right)=\\ =-\varepsilon_{0}\rho \cdot \left\{zr^{kinet}{\left(\Delta -\frac{r^{therm}}{\rho }-\frac{1/r^{therm}}{z}\right)}^{2}+\frac{\rho }{2r^{kinet}}{\left(\Delta -\frac{2r^{therm}}{\rho }\right)}^{2}\right\} \end{array}$$

Note that the time derivative of the free energy, generated by drift terms, is always negative in the absence of stirring (as it should be the case!).

However, whether this decrease in the Gibbs free energy leads eventually to the disappearance of the cluster or to its growth, is determined by the ratio of kinetic coefficients. Within some array of states, the fate of a nucleus is not determined **only** by thermodynamics. Instead, it depends **also** on the ratio of kinetic coefficients
$$\;r^{kinet}\equiv \sqrt{\frac{L_{D}}{L_{l}}},$$
see Figure 3. In other words, thermodynamic and kinetic criteria of critical nuclei may not coincide for anisotropic nucleation (more correctly: thermodynamic information–dependence of Gibbs free energy on both sizes, in case of regions III and IV, is not enough to say whether the cluster is subcritical or overcritical). Since this statement sounds radical (though actually being not so radical), let us try to formulate it more explicitly. 

Standard explanation in physics course tells us that thermodynamics dictates the general direction of evolution and kinetics determines the rate of evolution and, most probably, the choice of evolution path. This picture is absolutely correct if the state of the nucleus is determined by only a single parameter. If not, then the choice of evolution path is also the responsibility of kinetics. In other words, there may exist some “hidden kinetic parameters” which determine the direction of the evolution path together with thermodynamics. Many attempts have been suggested to increase the role of thermodynamics in determining the choice of evolution path. The most popular hypothesis is the maximum of Gibbs free energy decrease rate but it was never proved. Popular statements of the minimum of entropy production do not have a direct connection with the choice of the path—instead, it determines the final state of the path, which is equilibrium with zero entropy production in closed systems and steady-state with minimum entropy production in open systems. Moreover, even this rather limited statement is proven by Prigogine et al. to hold only for the case of constant kinetic coefficients [20]. 

In CNT for spherical or other clusters with an optimized surface (described by a single size parameter), if one has some cluster and at some moment “switches off” the random walk (the noise) in size space, one can predict the fate of the cluster (it either dissolves or grows) independently on any specific properties of the kinetic coefficients. In our case, for regions III and IV, this is not the case—without noise, clusters in these regions may grow eventually or shrink eventually, depending on kinetic coefficients. (We emphasize that in ALL four regions the evolution without noise always leads to a DECREASE in Gibbs free energy, as shown above.) Note that the discussed effects are different as compared with the deviation of the real evolution path from the optimal path along the steepest path over Gibbs free energy surface (and the corresponding deviation from the saddle-point) due to stochastic effects. The discussed here peculiar effects occur already in cases if only the drift terms are accounted for as it is done in the present study.

A possible difference between thermodynamically critical size and size with zero drift term during nucleation in binary alloys with large differences of diffusivities between the cluster and surrounding matrix was discussed in [21]. 

#### 3.1.3. Anisotropic Nucleation under Stirring

Now, let us consider nucleation criteria under stirring. Nucleation in open systems is known from other fields–for example, nucleation in the contact zone of reacting materials (in a sharp concentration gradient)—when thermodynamically stable (in closed system) nuclei can be kinetically suppressed in an open system (in reaction and diffusion zone) by the fast-growing neighboring compound layers [19]. Like in the previous subsection, let us “switch off” for the moment the noise terms in the equations governing nucleation. In such case, drift terms are determined by Equation (7), which we reformulate in the following form: (24)$$\begin{array}{l} \frac{d\rho }{d\tau }=r^{kinet}\Delta \left(1-\frac{r^{therm}}{\rho \Delta }-\frac{1/r^{therm}}{z\Delta }-\frac{r^{bal}}{r^{kinet}}\frac{J}{\Delta }\right)=r^{kinet}\Delta \left(1-b_{D}j-\xi -\eta \right)\\ \frac{dz}{d\tau }=\frac{1}{r^{kinet}}\Delta \left(1-b_{l}j-2\xi \right) \end{array}$$

Here
$$\xi \equiv \frac{1}{\Delta }\frac{r^{therm}}{\rho },\;\eta \equiv \frac{1}{\Delta }\frac{1/r^{therm}}{z}\;\Delta \equiv x-x_{eq},\;j=\frac{J}{\Delta },\;b_{D}\equiv \frac{r^{bal}}{r^{kinet}},\;b_{l}\equiv \frac{r^{kinet}}{r^{bal}}=\frac{1}{b_{D}}\;$$

In this paper, we consider only the case $\;b_{D}>b_{l}\;\left(r^{bal}/r^{kinet}>1\right)$ leading to elongated structures under stirring. The alternative case $\;b_{D}<b_{l}\;\left(r^{bal}/r^{kinet}<1\right)$ of disc-like structures will be considered elsewhere.

Note that in the symmetrical case $\;b_{D}=b_{l}\;\left(r^{bal}/r^{kinet}=1\right)$ an effective Gibbs free energy could be introduced instead of Equation (18) having the form:(25)$$\begin{array}{l} G^{ef}\left(\rho ,z|J\right)=G\left(\rho ,z\right)-\frac{1}{2}\varepsilon_{0}J\rho^{2}z=\\ =\varepsilon_{0}\cdot \left\{\frac{\rho^{2}\left(1-r^{therm}z\cdot \Delta \right)}{2r^{therm}}+2r^{therm}\rho z-\frac{1}{2}J\rho^{2}z\right\}=\\ =\varepsilon_{0}\cdot \left\{\frac{\rho^{2}\left(1-r^{therm}z\cdot \left(\Delta -J\right)\right)}{2r^{therm}}+2r^{therm}\rho z\right\} \end{array}$$
so that the analysis is reduced to the definition of an effective supersaturation Δ−J. (In the case of high enough ballistic factor *J*, J>Δ, all particles are dissolved. In our anisotropic case under stirring one cannot use the notion of effective Gibbs free energy).

The critical (stationary) state (both time derivatives are equal to zero):(26)$$\begin{array}{l} \begin{array}{l} \frac{dD}{dt}=0\Leftrightarrow \xi +\eta =1-b_{D}j\\ \frac{dl}{dt}=0\Leftrightarrow \;\xi =\frac{1}{2}-\frac{b_{l}}{2}j \end{array}\}\Rightarrow \\ \Rightarrow \xi^{crit}=\frac{1}{2}-\frac{b_{l}}{2}j,\;\eta^{crit}=\frac{1}{2}-\left(b_{D}-\frac{b_{l}}{2}\right)j:\;\\ \frac{r^{therm}}{\rho^{crit}}=\frac{\Delta }{2}-\frac{b_{l}}{2}J,\;\frac{1/r^{therm}}{z^{crit}}=\frac{\Delta }{2}-\left(b_{D}-\frac{b_{l}}{2}\right)J \end{array}$$

Thus, under stirring (in an open system), instead of Wulff’s rule for the equilibrium shape, one has
(27)$$\;\frac{r^{therm}}{\rho^{crit}}-\frac{1/r^{therm}}{z^{crit}}=\left(b_{D}-b_{l}\right)J=\left(\frac{r^{bal}}{r^{kinet}}-\frac{r^{kinet}}{r^{bal}}\right)J$$

As can be expected, in open systems the Wulff rule is violated even for critical nuclei. Here, depending on stirring intensity, one can distinguish two cases (Figure 4 and Figure 5).

Case 1:(28)$$1-b_{D}j>\frac{1}{2}-\frac{b_{l}}{2}j\Leftrightarrow j<\frac{1}{2b_{D}-b_{l}}$$

Case 2:(29)$$0<1-b_{D}j<\frac{1}{2}-\frac{b_{l}}{2}j\Leftrightarrow \frac{1}{b_{D}}>j>\frac{1}{2b_{D}-b_{l}}$$

In this case, a critical point in its usual meaning (both drift velocities simultaneously are equal to zero) does not exist. Three regions, instead of four above, exist: Region 1 of inevitable (at any ratio of kinetic factors) growth in both directions is separated from region 2 of inevitable (at any ratio of kinetic factors) shrinking by the region, in which the final direction of evolution is determined not only by thermodynamics (Gibbs free energy and its dependence on two sizes) but as well by the ratio of kinetic factors.

Region IV contains the states, for which the further evolution of the nucleus is determined not only by thermodynamics. Instead, the knowledge of the ratio of kinetic coefficients is needed, to predict the direction of evolution, $\;r^{kinet}\equiv \sqrt{\frac{L_{R}}{L_{l}}}$—see Figure 6.

Here we considered homogeneous nucleation processes. Most probably, the real nucleation process is heterogeneous, and the initial nuclei nucleate at some special facets of the preexisting particles [2]. In any case, the basic features concerning the possible fate of sufficiently large aggregates elaborated in the present analysis are expected to remain the same. Models of heterogeneous anisotropic nucleation at non-ideal facets under stirring will be considered elsewhere. 

### 3.2. Growth Stage (at Constant Supersaturation)

In this subsection, we analyze the growth and shape evolution of individual cylinders under stirring at constant supersaturation by numerical integration of Equation (7). We choose initial sizes within region II and trace the change of both sizes and aspect ratio with time, at various stirring intensities. The results are presented in Figure 7.

Asymptotically, if both sizes tend to infinity, one gets
(30)$$\begin{array}{l} \frac{d\rho }{d\tau }=r^{kinet}\left(\Delta -\frac{r^{therm}}{\rho }-\frac{1/r^{therm}}{z}-\frac{r^{bal}}{r^{kinet}}J\right)\approx r^{kinet}\Delta -r^{bal}J\\ \frac{dz}{d\tau }=\frac{1}{r^{kinet}}\left(\Delta -\frac{2r^{therm}}{\rho }-\frac{r^{kinet}}{r^{bal}}J\right)\approx \frac{\Delta }{r^{kinet}}-\frac{J}{r^{bal}} \end{array}$$
so that for very long annealing times (if supersaturation is kept constant)
(31)$$\frac{d\left(r^{therm}z-\rho /r^{therm}\right)}{d\tau }\approx \left(\frac{r^{therm}}{r^{kinet}}-\frac{r^{kinet}}{r^{therm}}\right)\Delta +\left(\frac{r^{bal}}{r^{therm}}-\frac{r^{therm}}{r^{bal}}\right)J$$
(32)$$\frac{z}{\rho }\to \frac{\frac{\Delta }{r^{kinet}}-\frac{J}{r^{bal}}}{r^{kinet}\Delta -r^{bal}J}={\left(r^{kinet}\right)}^{-2}\frac{\Delta -\frac{r^{kinet}}{r^{bal}}J}{\Delta -\frac{r^{bal}}{r^{kinet}}J}$$

For the parameters used in Figure 7, we obtain asymptotic values of the aspect ratio equal to 2.25, 4 and 7.75, which practically coincide with the results of numerical calculations shown in Figure 7.

Thus, at this stage, the asymmetry appears as a competition between two factors–asymmetry of thermally activated attachment/detachment kinetics and ballistic detachments at different facets. From Equation (30) one may see that the growth rate of the length tends to zero if the supersaturation tends to $\Delta_{2}=\frac{r^{kinet}}{r^{bal}}J$, and the growth rate of diameter tends to zero if $\Delta \to \Delta_{1}=\frac{r^{bal}}{r^{kinet}}J$. In this paper, we treat the case $\frac{r^{bal}}{r^{kinet}}>1$, so that $\Delta_{1}>\Delta_{2}$. So, one may expect that at the ripening stage, the system will choose the larger asymptotic supersaturation $\Delta_{1}=\frac{r^{bal}}{r^{kinet}}J=b_{D}J$. Otherwise (if $\Delta <=\frac{r^{bal}}{r^{kinet}}J$), according to the first of Equation (30), diameters of all rods will shrink up to their complete dissolution which is physically impossible at the growth and ripening stage. Instead, when $\Delta \to \Delta_{1}=\frac{r^{bal}}{r^{kinet}}J$, the growth rate of mean length tends to a constant, and the growth rate of mean diameter decreases but remains positive. Therefore, as we will see in the next Section 3.3, with an account of supersaturation decrease, the aspect ratio should grow much more rapidly. Note that for this one does not need the ballistic erosion in one direction much larger than in another: at the ripening stage, even a small excess of the ballistic ratio $r^{bal}\equiv \sqrt{\frac{U_{D}}{U_{L}}}$ over the kinetic one $r^{kinet}\equiv \sqrt{\frac{L_{D}}{L_{l}}}$ may lead to an unlimited (in time) increase in the shape parameter (but the rate of this growth, of course, depends on the magnitude of the excess). This effect will be studied in the next subsection.

### 3.3. Asymptotic Ripening Stage

Here we concentrate on the asymptotic ripening stage of an almost constant volume of the cluster phase which is only redistributed among different fibers. In conventionally treated ripening processes in closed systems, the supersaturation in the surrounding medium tends to zero, so that nucleation of new particles becomes impossible. As a consequence, the number, *N(t)*, of remaining particles decreases. (Finally, “Only one will remain” according to “Highlander” (https://highlander.fandom.com/wiki/There_can_be_only_one), but such a situation is beyond statistical approach.) 

Under stirring, one may assume that initially, just after switching on the stirring, the powder particles should at least partially dissolve, and supersaturation should increase. Asymptotically, the total volume of the parent phase approaches some constant value, supersaturation also tends to some constant, but this constant is not equal to zero, contrary to the case of commonly treated ripening without stirring. Despite non-zero supersaturation, nucleation of the new particles at the ripening stage can be neglected since the critical size under stirring (and the corresponding nucleation barrier) is increased by stirring (see Equation (16)).

In this paper, we consider analytically only the asymptotic behavior of ripening under stirring. Due to continuous growth of length at non-zero stirring, one should be careful with typical (for the asymptotic stage) simplifications. 

Let
$$b_{D}\equiv \frac{r^{bal}}{r^{kinet}}>1,\;b_{l}=1/b_{D}\equiv \frac{r^{kinet}}{r^{bal}}<1,\;J>0,\;t\to \infty$$

In this case, one has a tendency for rods to grow (in the opposite case, one should expect the discs to grow). Then, for a sufficiently advanced stage of ripening the supersaturation should tend to the maximum of two stirring factors $b_{D}J$, $b_{l}J$. Indeed, otherwise (if supersaturation is less than $b_{D}J$) all diameter growth rates are negative, so that diameters of all cylinders will shrink to zero, and the ripening problem loses sense. Thus, $\Delta \to b_{D}J\equiv \frac{r^{bal}}{r^{kinet}}J$. Then, one obtains
(33)$$\begin{array}{l} \frac{d\rho }{d\tau }=r^{kinet}\left(\left(\Delta -b_{D}J\right)-\frac{r^{therm}}{\rho }-\frac{1/r^{therm}}{z}\right)\\ \frac{dz}{d\tau }=\frac{1}{r^{kinet}}\left(\left(b_{D}-b_{l}\right)J-\frac{2r^{therm}}{\rho }\right) \end{array}$$

The second of these equations leads to the conclusion that at  t→∞ one expects
$$\left(b_{D}-b_{l}\right)J>>\frac{2r^{therm}}{<\rho >}$$
so that
(34)$$\begin{array}{l} \frac{dz}{d\tau }\approx \frac{1}{r^{kinet}}\left(\left(b_{D}-b_{l}\right)J\right)\approx const\;at\;t\to \infty \Rightarrow \\ \Rightarrow z_{i}\left(t\right)\approx z_{i0}+\frac{b_{D}-b_{l}}{r^{kinet}}J\tau \Rightarrow <z>\approx \frac{b_{D}-b_{l}}{r^{kinet}}J\tau \equiv V\tau \;at\;t\to \infty \end{array}$$

Thus, at the asymptotic stage, the distribution of rod lengths will shift with constant velocity and with almost constant shape (formed at the transient stage) and without further broadening. Thus, at a very late stage <*z*> tends to infinity much faster than the diameter. Consequently, we arrive at
(35)$$\begin{array}{l} \frac{d\rho }{d\tau }\approx r^{kinet}\left(\Delta^{\prime }-\frac{r^{therm}}{\rho }\right)\\ \Delta^{\prime }\equiv \left(\Delta -b_{D}J\right) \end{array}$$

This equation is similar to the kinetic equation for grain growth [22] suggested by Hillert in analogy to the LSW model for ripening [23,24]. It should lead to parabolic growth of the mean diameter
(36)$$<\rho {>}^{2}∼\tau ,\Delta^{\prime }\equiv \left(\Delta -b_{D}J\right)∼\frac{1}{\sqrt{\tau }}$$

The Hillert model employs the constraint of the constant total area in 2D grain structure ($\sum_{i}^{N\left(t\right)}R_{i}^{2}\approx const$) or constant total volume ($\sum_{i}^{N\left(t\right)}R_{i}^{3}\approx const$). In our case, we also use the constraint of almost constant volume but for another geometry (rods instead of spheres) and with mean length growing with a time faster than the diameter ($\sum_{i}^{N\left(t\right)}\rho_{i}^{2}z_{i}\approx const$). Therefore, one might expect that the asymptotic size distribution will differ from the Hillert distribution of diameters:(37)$$g^{Hillert}\left(\rho \right)=const\cdot \frac{{\left(\rho /\rho_{crit}\right)}^{2}}{{\left(2-\rho /\rho_{crit}\right)}^{4}}\exp\left(\frac{-4\rho /\rho_{crit}}{2-\rho /\rho_{crit}}\right)$$

Let us check this by explicit substitution. 

Equations (34) and (35) imply that one can expect the size distribution function to have the multiplicative form: (38)$$f\left(\rho ,z,\tau \right)=f_{\rho }\left(\rho ,\tau \right)\varphi \left(z-V\tau \right),\;\mathrm{with}\;V=\frac{b_{D}-b_{l}}{r^{kinet}}J.$$

Here $f_{\rho }\left(\rho ,\tau \right)d\rho$ is the number of cylinders per unit volume with diameters within the interval (ρ,ρ+dρ), φ(z)dz is a probability for any cylinder to have a length within the interval (*z*, *z* + *dz*). At that, $n\left(\tau \right)=\int_{0}^{\infty }f_{\rho }\left(\rho ,\tau \right)d\rho$ is the number of cylinders per unit volume, i.e.,
(39)$$1=\int_{V\tau }^{\infty }\varphi \left(z-V\tau \right)dz$$

The total volume of all nanobelts per unit volume should tend to a constant,
(40)$$\begin{array}{l} \int_{0}^{\infty }\frac{\pi }{4}\rho^{2}f_{\rho }\left(\rho ,\tau \right)d\rho \int_{V\tau }^{\infty }z\varphi \left(z-V\tau \right)dz=\int_{0}^{\infty }\frac{\pi }{4}\rho^{2}f_{\rho }\left(\rho ,\tau \right)d\rho \cdot \bar{z}\to const\Rightarrow \\ \Rightarrow \int_{0}^{\infty }\rho^{2}\frac{\partial }{\partial \tau }\left(f_{\rho }\left(\rho ,\tau \right)\right)d\rho \cdot \bar{z}+\int_{0}^{\infty }\rho^{2}f_{\rho }\left(\rho ,\tau \right)d\rho \cdot \frac{d\bar{z}}{d\tau }=0\Rightarrow \\ \Rightarrow -\int_{0}^{\infty }\rho^{2}\frac{\partial }{\partial \rho }\left(r^{kinet}\left(\Delta^{\prime }-\frac{r^{therm}}{\rho }\right)f_{\rho }\left(\rho ,\tau \right)\right)d\rho \cdot V\tau +\int_{0}^{\infty }f_{\rho }\left(\rho ,\tau \right)d\rho \cdot <\rho^{2}>V=0\Rightarrow \\ \Rightarrow -\int_{0}^{\infty }\frac{\partial }{\partial \rho }\left(\rho^{2}\left(r^{kinet}\left(\Delta^{\prime }-\frac{r^{therm}}{\rho }\right)f_{\rho }\left(\rho ,\tau \right)\right)\right)d\rho \cdot \tau +\int_{0}^{\infty }2\rho \left(r^{kinet}\left(\Delta^{\prime }-\frac{r^{therm}}{\rho }\right)f_{\rho }\left(\rho ,\tau \right)\right)d\rho \cdot \tau +\\ +\int_{0}^{\infty }f_{\rho }\left(\rho ,\tau \right)d\rho \cdot <\rho^{2}>=0\Rightarrow \\ \Rightarrow 0+2r^{kinet}\Delta^{\prime }<\rho >\tau -2r^{kinet}r^{therm}\tau +<\rho^{2}>=0\Rightarrow \\ \Rightarrow \Delta^{\prime }\equiv \Delta -b_{D}J=\frac{r^{therm}}{<\rho >}-\frac{<\rho^{2}>}{2r^{kinet}<\rho >\tau } \end{array}$$

Thus,
(41)$$\begin{array}{l} \frac{d\rho }{d\tau }\approx r^{kinet}\left(\frac{r^{therm}}{<\rho >}-\frac{<\rho^{2}>}{2r^{kinet}<\rho >\tau }-\frac{r^{therm}}{\rho }\right)=\\ =r^{kinet}r^{therm}\left(\frac{1}{<\rho >}\left(1-\frac{<\rho^{2}>}{2r^{kinet}r^{therm}\tau }\right)-\frac{1}{\rho }\right) \end{array}$$

Thus, at asymptotic stage the distribution over diameters satisfies the following approximate equation:(42)$$\begin{array}{l} \frac{\partial f_{\rho }}{\partial \tau }=-\frac{\partial }{\partial \rho }\left(r^{kinet}r^{therm}\left(\frac{1}{<\rho >}\left(1-\frac{<\rho^{2}>}{2r^{kinet}r^{therm}\tau }\right)-\frac{1}{\rho }\right)f_{\rho }\right)=\\ =-r^{kinet}r^{therm}\frac{\partial }{\partial \rho }\left(\left(\frac{1}{\rho_{crit}}-\frac{1}{\rho }\right)f_{\rho }\right) \end{array}$$

Here the critical diameter is determined as
(43)$$\rho_{crit}=\frac{<\rho >}{\left(1-\frac{<\rho^{2}>}{2r^{kinet}r^{therm}\tau }\right)}$$

We may look for a solution to Equation (42) in the form
(44)$$f_{\rho }=\Psi \left(\tau \right)g\left(\frac{\rho }{\rho_{crit}}\right),\;\int_{0}^{\infty }g\left(x\right)dx=1$$
(45)$$\Psi \left(\tau \right)=\frac{n\left(\tau \right)}{\rho_{crit}\left(\tau \right)}$$

Here, n(τ) is the number of rods per unit volume. Its product with mean squared diameter ($<\rho {>}^{2}∼\tau$) and mean length (<z>∼τ) should be asymptotically constant at the ripening stage. It means that
(46)$$n\left(\tau \right)∼\frac{const}{<\rho^{2}><z>}∼\tau^{-2}$$

Then, according to Equation (45),
(47)$$\Psi \left(\tau \right)=const\cdot \tau^{-5/2}$$

The solution of Equation (42) of the type (44) with volume constraint in the form (46) is found by standard but long algebra. (We used the math developed in [14], but one may follow also the widely known logic of the classical LSW papers or book [12]). In any case, anyone can check this solution by direct substitution:(48)$$\rho_{crit}=\sqrt{\frac{1}{2}r^{kin}r^{therm}\tau }$$
(49)$$\begin{array}{l} g\left(x=\rho /\rho_{crit}\right)=const\cdot \frac{{\left(\rho /\rho_{crit}\right)}^{1}}{{\left(2-\rho /\rho_{crit}\right)}^{6}}\exp\left(\frac{-4\rho /\rho_{crit}}{2-\rho /\rho_{crit}}\right),\\ <x>=\frac{13}{16},\;<x^{2}>=\frac{3}{4},\;<x>=1-\frac{1}{4}<x^{2}>\\ \sqrt{<\rho^{2}>}=\frac{\sqrt{3}}{2}\rho_{crit}=\sqrt{\frac{3}{8}r^{kinet}r^{therm}\tau } \end{array}$$

One may see that Equation (49) contains the same exponent as in the Hillert distribution (37), but the powers of factors before exponent differ—see Figure 8.

Thus, in the asymptotic region of long stirring, the mean length is proportional to time, and the mean diameter is proportional to the square root of time. It means that at the ripening stage of the intensive stirring, the aspect ratio may grow without limits. 

## 4. Discussion and Conclusions

Processes of redistribution between different particles and between different facets of the same particle are very complicated. In this paper, we consider only the simplest approximation of such redistribution (adding ballistic detachment rates). Such modification, in the isotropic case, would just redefine an effective supersaturation. In anisotropic case, when ballistic terms are different at different facets, they lead to much more interesting behavior.At the nucleation stage, even if stochastic terms are switched off, the size space contains three regions: (1) absolutely unstable (embryo shrinks in volume to zero at any kinetic coefficients), (2) absolutely stable (nuclei increase their volume at any kinetic coefficients) and (3) transient region, in which the fate of the particle is determined but the set of kinetic coefficients.Stirring shifts the boundaries of these regions and even may make the nucleation impossible.At the growth stage, one may suggest at least two possible reasons for anisotropic growth of elongated particles: (1) large anisotropy of the kinetic factors of growth/dissolution in two directions (*r*^therm^ << 1, *L_D_* << *L_l_*), (2) moderate anisotropy of ballistic terms (*r*^bal^/*r*^therm^ > 1, *U_D_*/*L_D_* > *U_l_*/*L_l_*). In this paper, we concentrate on the analysis of the second possibility. We demonstrate that this possibility provides results that are qualitatively similar to some experimental results [1,2] (large mean aspect ratio l/*D* after a long stirring period and approximately linear dependence of mean length on the stirring intensity, also see item 5). Under stirring, at the growth stage, both sizes grow with time so that the aspect ratio tends to a constant determined by Equation (32) depending on stirring intensity.At advanced ripening stages, the mean length tends to infinity linearly with time (the growth rate being proportional to ballistic factor, *J*), the mean diameter grows parabolically, and the aspect ratio tends to infinity.Asymptotic diameter distribution satisfies the modified Hillert relation (49).In future work, one should pay special attention to anisotropic heterogeneous nucleation at the facets under stirring. This is a part of the general problem of flow-induced crystallization that is known in polymer science [25,26].

## Figures and Tables

**Figure 1 entropy-22-01254-f001:**
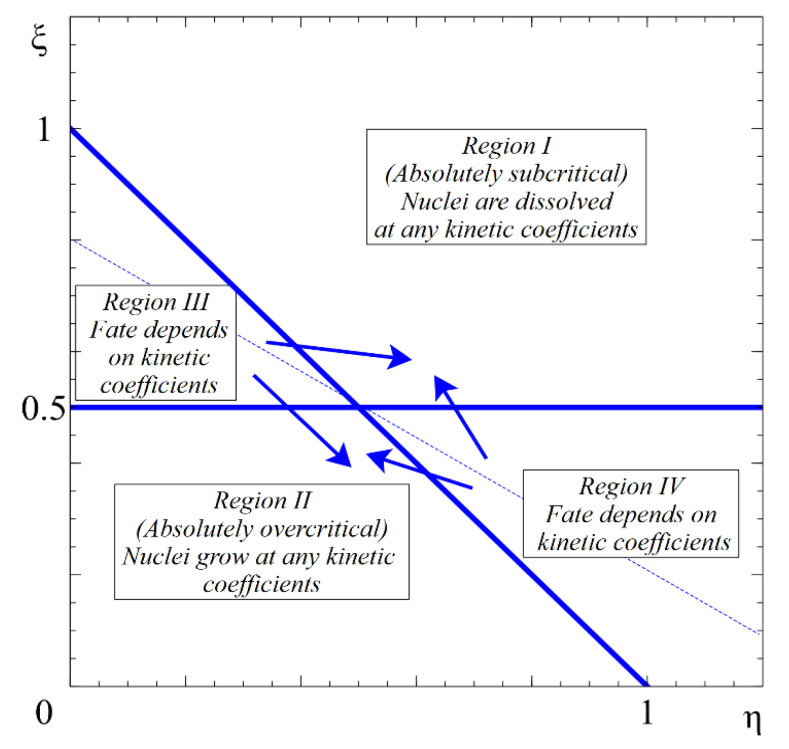
Anisotropic nucleation regimes (without stirring). The fate of any nucleus in region I consists in further shrinking to full disappearance, at any combination of kinetic coefficients. The fate of any nucleus in region II is further growth, also at any combination of kinetic coefficients. The fate of nuclei in regions III and IV is defined by the ratio of kinetic coefficients $\;r^{kinet}\equiv \sqrt{\frac{L_{D}}{L_{l}}}$—see Figure 2.

**Figure 2 entropy-22-01254-f002:**
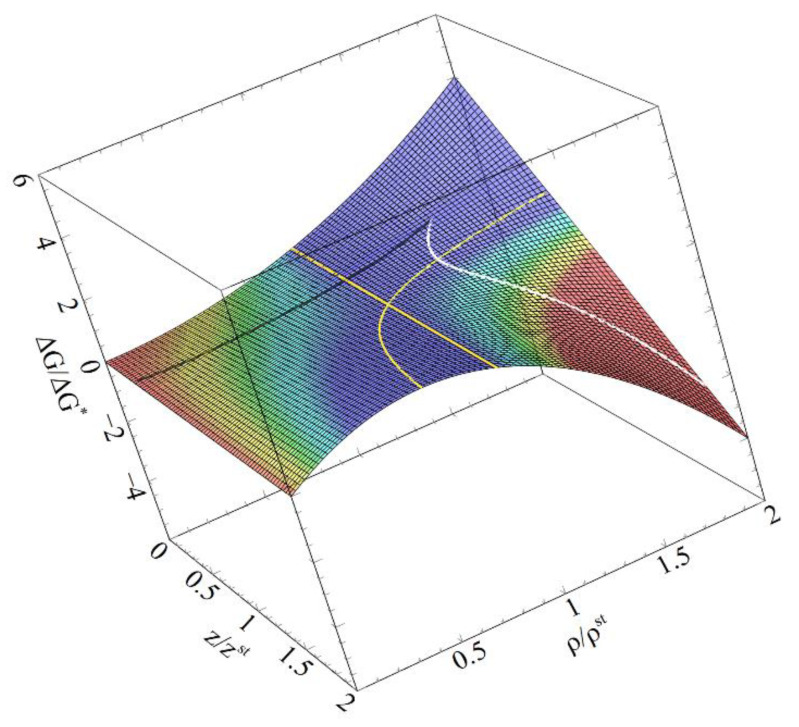
Surface of reduced Gibbs free energy change, ΔG/ΔG*, as a function of reduced sizes, *z*/*z*^st^, and *ρ*/*ρ*^st^. Downhill paths of evolution (characterized by a decrease in the Gibbs free energy) may be realized in region IV by either growth or shrinkage dependant on the kinetic factor (in our case *r^kinet^* = 1 and 2).

**Figure 3 entropy-22-01254-f003:**
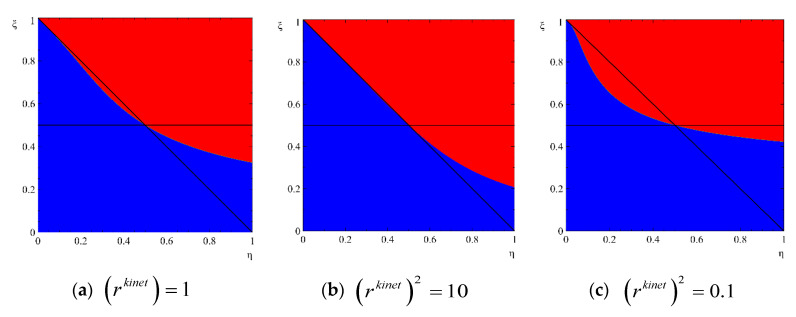
Anisotropic nucleation regimes without stirring (J = 0) at various ratios of kinetic coefficients $\;r^{kinet}\equiv \sqrt{\frac{L_{R}}{L_{l}}}$. Red color corresponds to states leading to shrinking, blue color corresponds to states leading to irreversible growth. (**a**) $\left(r^{kinet}\right)=1$, (**b**) ${\left(r^{kinet}\right)}^{2}=10$, (**c**) $\;{\left(r^{kinet}\right)}^{2}=0.1$.

**Figure 4 entropy-22-01254-f004:**
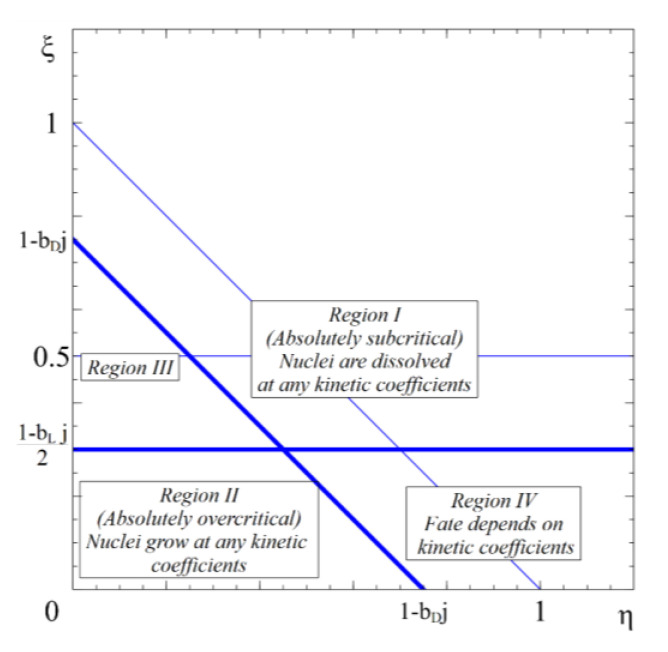
Anisotropic nucleation regimes under stirring of “small” intensity: $j\equiv \frac{J}{\Delta }<\frac{1}{2b_{D}-b_{l}}$. The fate of any nucleus in region I is further shrinking to full disappearance, at any combination of kinetic coefficients. The fate of any nucleus in region II is further growth, also at any combination of kinetic coefficients. The fate of nuclei in regions III and IV depends on the ratio of the kinetic coefficients $\;r^{kinet}\equiv \sqrt{\frac{L_{R}}{L_{l}}}$.

**Figure 5 entropy-22-01254-f005:**
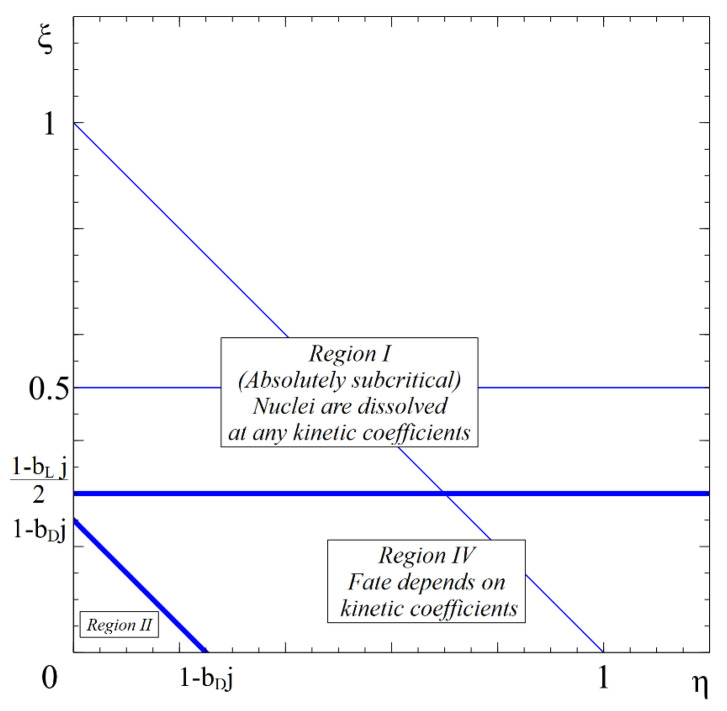
Anisotropic nucleation regimes under stirring of “large” intensity: $\frac{1}{b_{D}}>j\equiv \frac{J}{\Delta }>\frac{1}{2b_{D}-b_{l}}$. The fate of any nucleus in region I is further shrinking to full disappearance, at any combination of kinetic coefficients. Fate of any nucleus in region II is further growth, also at any combination of kinetic coefficients. Region III is absent (compare with Figure 4). Regions I and II are separated by region IV. The fate of nuclei in region IV depends on the ratio of kinetic coefficients $\;r^{kinet}\equiv \sqrt{\frac{L_{R}}{L_{l}}}$.

**Figure 6 entropy-22-01254-f006:**
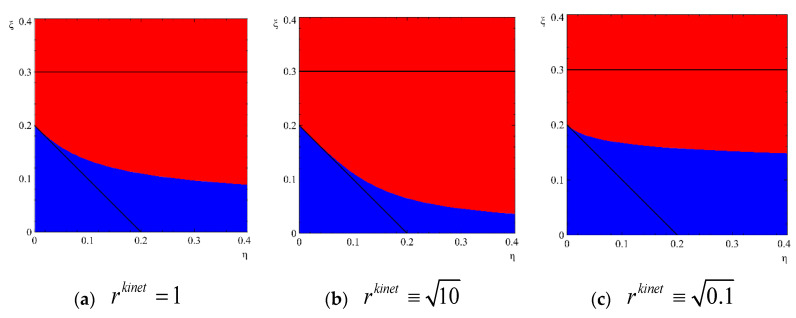
Anisotropic nucleation regimes under stirring (J = 0.04) at various ratios of kinetic coefficients, $\;r^{kinet}\equiv \sqrt{\frac{L_{R}}{L_{l}}}$. Red color corresponds to states leading to shrinking, blue color corresponds to states leading to irreversible growth. (**a**) $\left(r^{kinet}\right)=1$, (**b**) ${\left(r^{kinet}\right)}^{2}=10$, (**c**)$\;{\left(r^{kinet}\right)}^{2}=0.1$.

**Figure 7 entropy-22-01254-f007:**
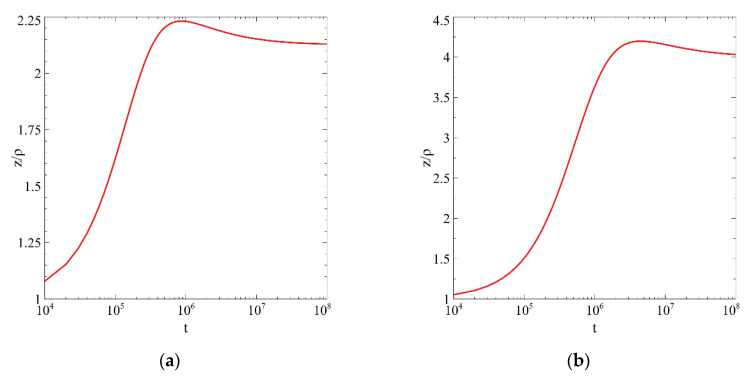
Time dependencies of aspect ratio at $\;r^{kinet}=1,r^{therm}=1,r^{bal}=2$ and various stirring intensity. J/Δ= 0.3 (**a**), 0.4 (**b**), 0.45 (**c**).

**Figure 8 entropy-22-01254-f008:**
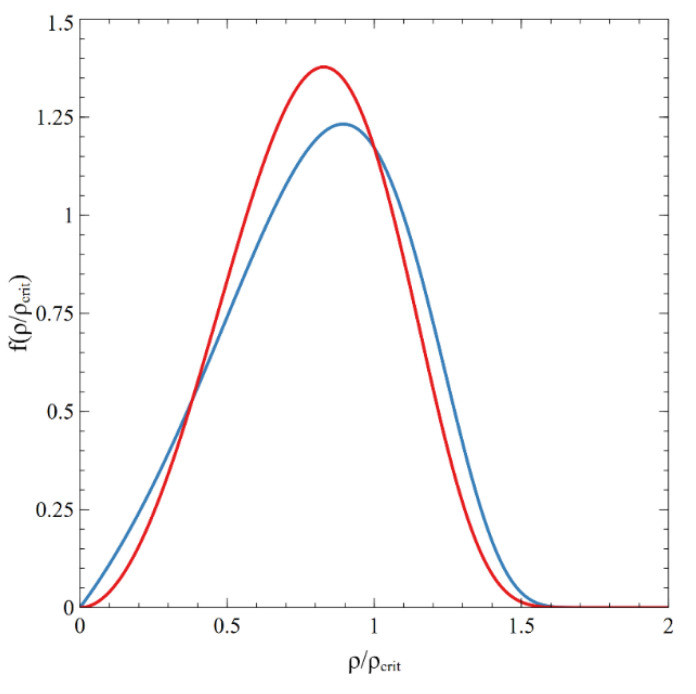
Comparison of normalized diameter distribution (blue line, Equation (49)) with Hillert distribution (red line, Equation (37)).

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
