# Peer review of "Anisotropic Nucleation, Growth and Ripening under Stirring—A Phenomenological Model"

_entropy, 2020, doi:10.3390/e22111254_

Round 1

Reviewer 1 Report

The manuscript by Gusak, Huriev and Schmelzer presents an interesting theoretical/numerical investigation of the anisotropic formation of elongated (cylindrical in their example) aggregates under conditions of intense stirring. They consistently extend the classical classical nucleation theory to open systems under stirring conditions. Their use of ballistic terms to analyze the influence of stirring on the drift terms is especially interesting and original. Overall, the paper is very clearly written, especially the first half, and easily understandable by an average reader. I only have a few minor remarks/clarifications and a more basic question.

- the names of the variables are introduced in a bit of a random manner, some of them, unless I am mistaken, are not defined at all but must be looked up in their previous publications. It would be quite useful to the reader if the meaning of the variables was presented in a systematic manner, including units.

- there is a weird superindex in Eq.2 "int erface".

- the factors of 2 in the same equation "account for ... from both sides", which sounds reasonable for the flat ends of the cylinder. But, what is exactly the meaning of "both sides" for the lateral area of the cylinder?

- l110: "The resulting in such approach non-linearities" sounds strange in English.

- l143-144: "a detailed Fokker-Planck approach will be considered elsewhere)": since the authors have all necessary ingredients (diffusion and drift terms) to set up and solve a FP equation, why do they not follow this approach? Perhaps they may care to elaborate on this in a bit more detail.

- l271: what is the exact meaning of "optimized surface"? "described by a single size parameter" hardly seems to be a definition of an optimized surface. Do they refer to a sphere? to minimal surfaces in general?

- l277: "the discussed here effects" sounds strange in English.

- l332-334: the denomination of Regions 1 and 2 as "unambiguous" suggests the other(s) are "ambiguous", i.e. "open to more than one interpretation" according to the dictionary, which is not the case. Growth in regions III (IV) is not ambiguous, it happens to be determined unambiguously by more than just thermodynamics.

- l403: the Highlander quotation is ok, why not, but should be accompanied by a literature reference. Some readers may not know what it is about.

- l421: "looses sense"

- l427-428: "the length distribution will be the early (and at transient stage) formed distribution" sounds strange in English.

The basic question I referred to above has to do with the subject matter of the manuscript and its adequacy in a journal like Entropy. The manuscript would clearly fit much better in a journal like Crystal (to mention another MDPI journal). The link between this manuscript and Entropy is a bit far-fetched. I would strongly suggest the authors submit their manuscript to Crystals, where, I am sure, it will be accepted with very minor revisions.

Reviewer 2 Report

The paper "Anisotropic Nucleation, Growth and Ripening under Stirring – a Phenomenological Model" is devoted to the important theme and presents very interesting results and presents very interesting results. It is well-written and clearly organized. I suggest to make a paper more readable, more understandable for a reader.

  1. Please specify under the revision the characteristic time and spatial scales inherent for the addressed physical processes.
  2. What are the dimensionless numbers relevant for understanding of the reported processes? What are their values?
  3. I think that the comparison of the introduced models with the reported experimental results will strengthen  the manuscript.
